# The Reliability of Iodine Concentration in Diaper-Retrieved Infant Urine Using Urine Collection Pads, and in Their Mothers’ Breastmilk

**DOI:** 10.3390/biom10020295

**Published:** 2020-02-13

**Authors:** Kjersti Sletten Bakken, Ingvild Oma, Synne Groufh-Jacobsen, Beate Stokke Solvik, Lise Mette Mosand, Mina Marthinsen Langfjord, Elin Lovise Folven Gjengedal, Sigrun Henjum, Tor Arne Strand

**Affiliations:** 1Women’s Clinic at Lillehammer Hospital, Innlandet Hospital Trust, 2629 Lillehammer, Norway; beate.stokke.solvik@sykehuset-innlandet.no; 2Department of Medical Microbiology, Innlandet Hospital Trust, 2629 Lillehammer, Norway; ingvild.oma@sykehuset-innlandet.no (I.O.); lise.mette.mosand@sykehuset-innlandet.no (L.M.M.); 3Department of Research, Innlandet Hospital Trust, 2629 Lillehammer, Norway; tors@me.com (S.G.-J.); synnegroufh@hotmail.com (T.A.S.); 4Faculty of Environmental Sciences and Natural Resource Management, Norwegian University of Life Sciences, 1432 Ås, Norway; mina.marthinsen.langfjord@nmbu.no (M.M.L.); elin.gjengedal@nmbu.no (E.L.F.G.); 5Department of Nursing and Health Promotion, Faculty of Health Sciences, OsloMet–Oslo Metropolitan University, 0130 Oslo, Norway; sigrun.henjum@oslomet.no

**Keywords:** urinary iodine concentration (UIC), breastmilk iodine concentration (BIC), intraindividual variability, infant

## Abstract

Mild to moderate iodine deficiency is common among women of childbearing age. Data on iodine status in infants are sparse, partly due to the challenges in collecting urine. Urinary iodine concentration (UIC) is considered a good marker for recent dietary iodine intake and status in populations. The aim of this study was to investigate the reliability of iodine concentration measured in two spot-samples from the same day of diaper-retrieved infant urine and in their mothers’ breastmilk. We collected urine and breastmilk from a sample of 27 infants and 25 mothers participating in a cross-sectional study at two public healthcare clinics in Norway. The reliability of iodine concentration was assessed by calculating the intraclass correlation coefficients (ICC) and the coefficient of variation (CV). The ICC for infants’ urine was 0.64 (95% confidence interval (CI) 0.36–0.82), while the ICC for breastmilk was 0.83 (95% CI 0.65–0.92) Similarly, the intraindividual CV for UIC was 0.25 and 0.14 for breastmilk iodine concentration (BIC). Compared to standard methods of collecting urine for measuring iodine concentration, the diaper-pad collection method does not substantially affect the reliability of the measurements.

## 1. Introduction

Iodine deficiency leads to inadequate thyroid hormone production, which is vital for energy metabolism and for the developing central nervous system, and is considered to be the single most important cause of preventable mental impairment [1,2]. Globally, iodine deficiency caused goiter in 187 million people (2.7%) in 2010 which resulted in 2700 deaths in 2013 [3]. Iodine deficiency is re-emerging in Europe, and recent findings from Italy showed that iodine deficiency is still an issue in Europe [4,5].

Urinary iodine concentration (UIC) is considered to be a good marker for recent dietary iodine intake and status since over 90% of dietary iodine is excreted in the urine [6,7]. Nevertheless, UIC varies substantially within individuals depending on their hydration status and/or iodine intake [8]. Therefore, UIC is useful as a population estimate of iodine status, but not as an individual estimate [9]. Even so, compared to the intraindividual difference, larger differences have been observed between individuals when multiple urine samples have been collected [10]. By adjusting for intraindividual variability with more than one spot-sample, the interpretability of UIC data could be improved [11]. Many other elements are excreted in the urine such as As, Pb, Zn, Se, Cd, and Cu [12]; the urine concentrations of these are comparable with that of iodine. For most of these elements, however, status is usually measured by estimating the plasma or serum concentrations.

Median UIC ≥100 μg/L is considered to reflect an adequate iodine intake for children under the age of two years [13]. Breastmilk iodine concentrations (BIC) have been found to correlate well with infants UICs and is a good indicator of infant iodine intake, especially those who are fully breastfed [14,15]. The iodine content of breastmilk varies with maternal iodine intake and breastfed infants are therefore highly sensitive to variations in maternal dietary iodine intake [16,17]. There is no established cut-off value for BIC, however, some studies have suggested that BIC ≥100 μg/L may be considered as adequate [16,17,18].

Collection of infant urine can be a great challenge and collecting urine for 24 h in infants in order to estimate their total iodine secretion is considered almost impossible and spot-sampling is a more feasible alternative. There are different methods for collection of infant urine including suprapubic aspiration, urethral catheterization, clean catch void, urine collection bag or urine collection pads [19]. The latter two methods are those employed when collecting urine in healthy infants. Urine collection bags are often easily detached from the skin, causing loss and/or contamination of the sample. A diaper-pad collection method may be feasible [20], however loss and contamination might occur affecting both the reliability and accuracy. We are not aware of any studies that have estimated the reliability of the method, other than having performed a contamination test [14,21]. The aim of this study was to investigate the intraindividual variability of iodine concentration measured in two spot-samples of diaper-retrieved infant urine and mother’s breastmilk in a sub-sample population from a cross-sectional study.

## 2. Materials and Methods

### 2.1. Collection of Breastmilk and Urine

In a cross-sectional study, we recruited participants from public healthcare clinics in the inland area of Norway. The mother-infant dyads were enrolled during scheduled visits to the public healthcare clinics. The mothers were informed about the study purpose on the day of recruitment and given necessary instructions for the collection of biological samples. A sample of 27 mother-infant dyads were recruited to estimate the intraindividual variability of the infant’s UIC and the mother’s BIC. Figure 1 shows the data collection procedure used for these analyses.

The mothers were instructed to collect one spot breastmilk sample (5 mL) in the morning before breastfeeding and a second spot breastmilk sample (5 mL) in the afternoon after breastfeeding (Figure 1). Furthermore, they were instructed to collect one spot urine sample (5 mL) from the infant in the morning and another one in the afternoon, using two Sterisets Urine collection packs 310,019 (Sterisets International BV, Oss The Netherlands) containing one syringe (5 mL), one specimen container (20 mL), two uricol collection pads (21 cm x 7 cm), and an instruction leaflet. The mothers placed a pad inside the disposable diaper and checked it every 5 min until the pad was wet with urine, but not soiled by feces. The pad was removed, laid flat and urine was extracted using a disposable syringe. The urine was then transferred to the sterile container. An alternative method for extracting the urine was also presented to them. This method involved cutting the pad open, putting the wet cotton into the opened syringe and pressing the urine into the container. Each mother donated two repeated spot breastmilk samples (2 × 5 mL) and two repeated spot infant urine samples (2 × 5 mL), on the same day once during the study period. The containers were kept refrigerated until the sample was transferred to a –70 °C freezer and later transported on dry-ice for analysis. The breastmilk was poured directly into containers (Sarstedt screw cap tube, 50 mL, 114 × 28 mm, polypropylene (PP), Sarstedt Nümbrecht, Germany) and handled in the same manner as the urine samples.

### 2.2. Chemical Analyses

The determination of UIC and BIC was performed at the Norwegian University of Life Sciences, Faculty of Environmental Sciences and Natural Resource management. The urine samples were thawed and aliquoted into 15 mL pp centrifuge tubes (Sarstedt, Nümbrecht, Germany) by means of a 100–5000 µL electronic pipette (Biohit, Helsinki, Finland). Specifically, an aliquot of 1.00 mL of urine was diluted to 10.0 mL with an alkaline mixture (BENT), containing 4% (weight (w)/volume (V)) 1-Butanol, 0.1% (*w*/*v*) H4EDTA, 2% (*w*/*v*) NH4OH, and 0.1% (*w*/*v*) TritonTM X-100 (Millipore, Burlington, MA, USA). The concentration of NH4OH in BENT was set to 2% to avoid precipitation of struvite (MgNH4PO4 x 6H2O) in the urine.

The breastmilk samples were thawed and heated to 37 °C in a heating cabinet, homogenized, and subsequently prepared by dilution in an alkaline solution (BENT, similar to the one used for the urine samples, except in this case the concentration of NH4OH in BENT was increased to 5% (*w*/*v*). A conformance test between volume and weight of breast milk confined the concentration of iodine to two significant figures.

Method blank samples and Standard Reference Materials (SRM) were prepared in the same manner as the respective sample matrices. Deionized water (>18 MΩ) and reagents of analytical grade or better were used throughout. The quantification of iodine was performed by means of an Agilent 8900 ICP-QQQ (Triple Quadrupole Inductively Coupled Plasma Mass Spectrometer; Agilent Technologies, Hachioji, Japan) using oxygen reaction mode. Iodine was quantified on mass 127. 129I was used for correction of non-spectral interferences.

Considering breastmilk, the limit of detection (LOD) was 0.04 µg/L and the limit of quantification (LOQ) was 0.14 µg/L. Regarding urine, LOD was 0.1 µg/L and LOQ was 0.32 µg/L. The LOD and LOQ were calculated by multiplying the standard deviation of five method blank samples that followed each treatment by three and ten, respectively. To ensure methodological traceability and to check for accuracy in the determinations of iodine in breast milk and urine, SRM were used. Allowing for a coverage factor k = 2, corresponding to a level of confidence of about 95%, our results were within the recommended values issued for the Seronorm™ (Oslo, Norway) Trace Elements Urine L-1, Seronorm^TM^ Trace Elements Urine L-2, and the European Reference Materials ERM^®^-BD 150 (Geel, Belgium) and ERM^®^ -BD 151 Skimmed milk powders. The measurement repeatability was 1.6% with respect to both breast milk and urine.

A performance test of the sampling method was conducted regarding potential (i) decrease in or (ii) increase in iodine in urine. An aliquot of 10 mL of spot urine was added to a Sterisets Urine collection pad. After two minutes the urine was extracted and transferred into a 15 mL pp centrifuge tube (Sarstedt, Nümbrecht, Germany) by means of the associated disposable syringe. Further on, an aliquot of 1.00 mL of urine was diluted to 10.0 mL with the alkaline mixture (BENT). The procedure was repeated on ten different collection pads. An additional four collection pads had 10 mL of deionized water added and were prepared in the same manner as the urine samples (method blanks). Finally, five aliquots of 1.00 mL urine withdrawn directly from the spot urine sample and five aliquots of 1.00 mL deionized water only, were diluted to 10.0 mL with BENT. The performance test showed no decrease or increase in analyte; thus, the sampling method can be considered reliable.

### 2.3. Statistical Analyses

Statistical analyses were performed using Stata/SE 15.1 (StataCorp, College Station, TX, USA). Differences in median iodine concentrations and intakes were tested using the Wilcoxon matched-pairs signed-ranks test. Intraclass correlation coefficients (ICC) and 95% confident intervals were calculated based on a mean-rating (k = 2), absolute-agreement, 2-way random-effects model. ICC reflects both a degree of correlation and agreement between measurements. Values less than 0.5 are indicative of poor reliability, values between 0.5 and 0.75 indicate moderate reliability, values between 0.75 and 0.9 indicate good reliability, and values greater than 0.90 indicate excellent reliability. Furthermore, we calculated the intraindividual coefficient of variation (CV) for each participants’ UIC and BIC by dividing the difference between the two spot samples with their means. The estimated overall mean intraindividual CV was then calculated by totaling all the CV values of participants and dividing them by the number of participants and expressed as CV%. Statistical significance was set at *p* < 0.05. In addition, plots displaying variation in iodine concentration between two spot samples were made using the package blandr in RStudio for Mac version 1.1.463 (RStudio: Integrated Development for R., RStudio, Inc., Boston, MA URL http://www.rstudio.com/).

### 2.4. Ethical Considerations

All mothers gave their informed consent for inclusion before they participated in the study with their infants. The mothers were rewarded with a gift card of 600 NOK. The study was conducted in accordance with the Declaration of Helsinki, and the protocol was approved by the Regional Committee for Medical and Health Research Ethics (REC South East; Ref. no. 2018/1230).

## 3. Results

Of the included mother-infant dyads, all except two infants where breastfed, and the majority (74%) of the infants were exclusively breastfed (Table 1). The mean age of the included infants were 20.6 weeks, with a range from 4 to 52 weeks. Most of the infants (85.2%) were breastfeed shortly before the spot urine sample was taken. All the included women in this sub-sample were born in Norway.

The median iodine concentration of the two spot-samples of urine in infants differed by 10 µg/L and in breastmilk the difference was 20 µg/L, as shown in Table 2.

The difference in UIC between the two spot samples ranged from 0–370 µg/L (Figure 2). The Wilcoxon matched-pairs signed-ranks test showed that the distributions between the two spot-samples of urine for the whole sample of infants were the same (*p* = 0.64).

For BIC the difference between the two samples ranged from 0–129 µg/L (Figure 3). The distribution of BIC was the same in the two samples of breastmilk for the whole sample population (*p* = 0.26).

The ICC for infant UIC was 0.64 (95% CI 0.36–0.82), indicating a poor to moderate agreement of iodine concentrations between the two spot samples. However, the mean CV for infant UIC was 25%, ranging from 0%–56%.

For BIC, the ICC was 0.83 (95% CI 0.65–0.92), indicating a moderate to good agreement of iodine concentrations in the two different samples. This is also reflected in the mean CV for BIC which was 14%, with a range of 0%–51%.

The participating mothers reported that the pad was easy to use. However, the extraction of urine could be somewhat difficult and several of the mothers used the presented alternative method. As they were supplied with disposable gloves at recruitment, they felt that this was an acceptable method for collecting the infant’s urine. Some mothers were given extra pads, as their infant had soiled the diaper with feces several times.

## 4. Discussion

In this analysis, we measured the variability of iodine concentration in two spot samples of diaper-retrieved infant urine and mother’s breastmilk among a sample of 27 infants and 25 mothers from a Norwegian cross-sectional study. We found that the ICC of infant UIC had a poor to moderate agreement. However, the mean intraindividual CV of UIC was found to be low, and the observed means of the two spot-samples of UIC were almost identical. In general, it was those with the highest mean iodine concentrations that had the highest variation between the two samples. This is comparable with the ICC observed in another Norwegian study using two morning spot urine samples collected using a urine collection bag in children from three years and older and in adults [22]. Furthermore, the mean CV in infant UIC at 25% was somewhat lower compared to a CV of 38% that König and colleagues found in Switzerland [23]. However, König et al. collected samples on ten different days, and we collected two spot samples on the same day, therefore, their somewhat higher mean CV also includes the day-to-day variation in iodine and fluid intake.

The variability, expressed by the ICCs and the mean CV, was somewhat higher for UIC than for BIC. This difference could be explained by the fact that the volume of breastmilk varies less than urine. However, it should be noted that there was a considerable overlap in the CIs of the ICCs. In addition, there is a higher fraction of iodine excreted in breastmilk compared to maternal urine [18]. Therefore, BIC reflects infant iodine intake better than UIC [18]. 

Iodine is an essential micronutrient that has to be under tight homeostatic control. Therefore, the body possesses a number of mechanisms by which it can absorb, concentrate and excrete iodine in the form of its monovalent anion iodide. This system encompasses several organ systems and different physiological processes, all to ensure that iodine processing and utilization take place properly [24]. In a multicenter study, Dold and colleagues found that in lactating women with low iodine intakes fractional iodine excretion in breastmilk increased and excretion in urine decreased [18]. Furthermore, they found that the proportion of iodine excretion was constant in breastmilk at 33% in mothers who were iodine deficient [18]. UIC varies throughout the day which could explain some of the variability observed in our study. However, in our sub-sample, the UIC in the two spot-samples did not increase from the morning sample to the afternoon sample, as previous studies have implied [25,26].

A laboratory performance test of the sampling method using the Sterisets Urine collection pack showed no decrease or increase in analyte (100% recovery); thus, in this respect, the sampling method can be considered reliable. We are not aware of any studies that have validated the use of a pad for the collection of urine for measuring infant UIC. Moreover, the participating mother’s overall experiences with using the pad for collection of infant urine was mostly positive, which underlines the feasibility of this method. In other words, it seems like diaper-retrieved infant urine for measuring UIC is not inferior to other collection methods.

## 5. Conclusions

The use of collection pads in disposable diapers offers a feasible method for measuring iodine status in infants, which does not substantially add variability to the measurements.

## Figures and Tables

**Figure 1 biomolecules-10-00295-f001:**
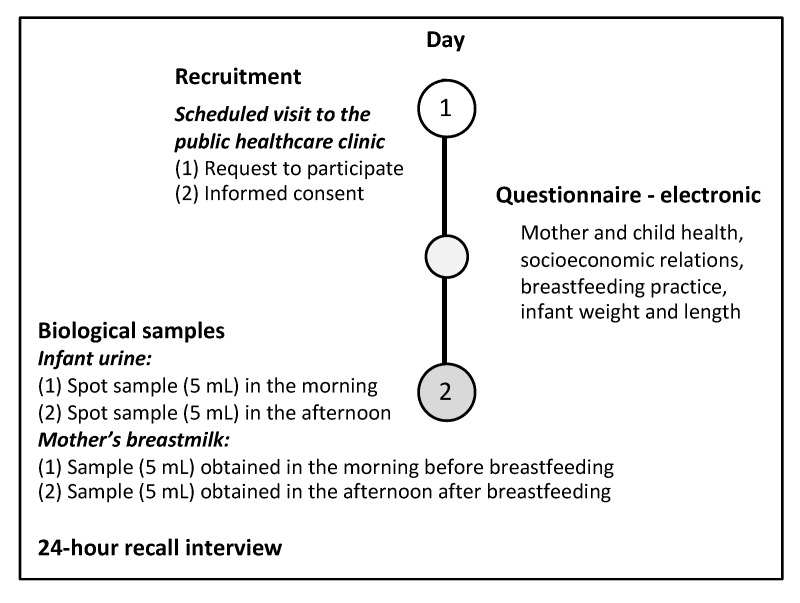
Data collection procedure.

**Figure 2 biomolecules-10-00295-f002:**
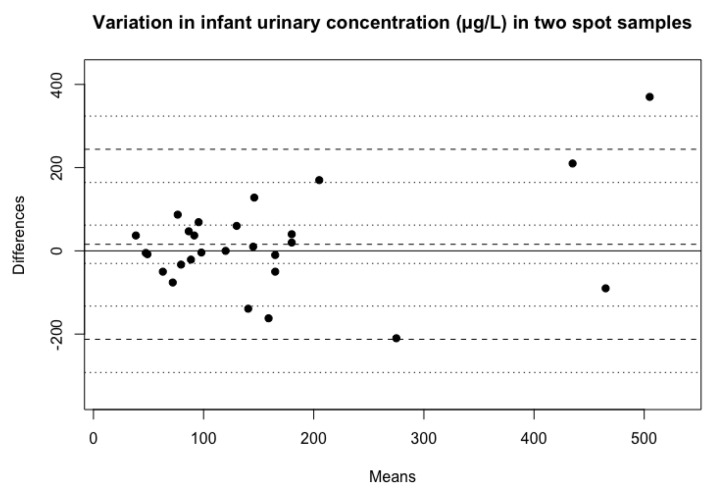
Variation in iodine concentration between the two spot samples of infant urine, including observed average agreement and 95% confidence intervals. y = 0 is the line of perfect average agreement.

**Figure 3 biomolecules-10-00295-f003:**
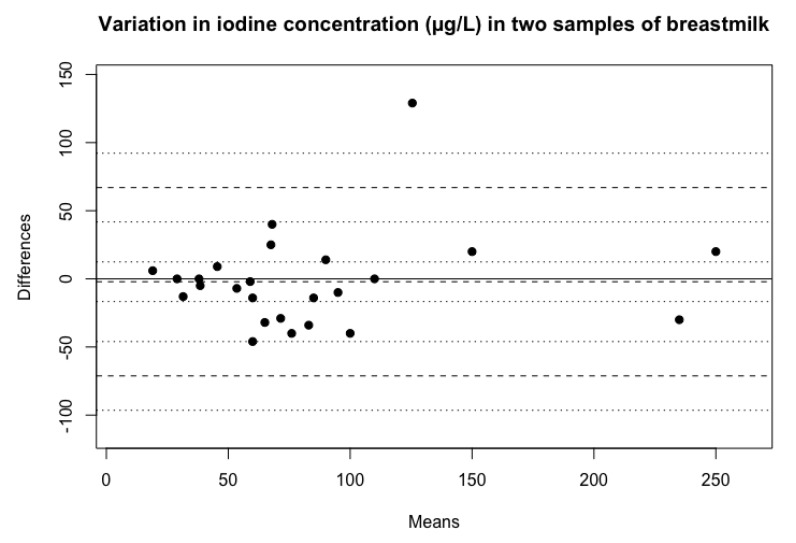
Variation in iodine concentration between two spot samples of breastmilk, including observed average agreement and 95% confidence intervals. y = 0 is the line of perfect average agreement.

**Table 1 biomolecules-10-00295-t001:** Characteristics of included mother-infant dyads *n* = 27.

	n (%) ^1^
Maternal daily smoking	1 (4)
Maternal education > 4 years of higher education	13 (48)
Infant age in weeks, median (min–max)	18 (4–52)
Exclusively breastfed infants	20 (74)
The infant was breastfed before urine-sample was taken	23 (85)
Maternal age in years, median (min–max)	30 (20–39)

^1^ Data are presented as number (n) and % unless indicated otherwise.

**Table 2 biomolecules-10-00295-t002:** Mean and median iodine concentration (µg/L) measured in two sets of samples from the infant’s urine (*n* = 27) and in mother’s breastmilk (*n* = 25).

	Mean (Standard Deviation)	Median (min–max)
Infant urinary iodine concentration (*n* = 27)		
Sample 1	167.2 (155.1)	120 (34–690)
Sample 2	151.4 (116.3)	110 (20–510)
Breastmilk iodine concentration (*n* = 25)		
Sample 1	83.4 (60.5)	62 (22–260)
Sample 2	92.4 (66.4)	82 (16–270)

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
