# Peer review of "The Reliability of Iodine Concentration in Diaper-Retrieved Infant Urine Using Urine Collection Pads, and in Their Mothers’ Breastmilk"

_biomolecules, 2020, doi:10.3390/biom10020295_

Round 1

Reviewer 1 Report

Dear authors,

I reviewed with interest the paper entitled “The reliability of iodine concentration in diaper-2 retrieved infant urine using urine collection pads, 3 and in their mothers’ breastmilk”.

The paper is well written and data are clearly presented. I would suggest to format the paper as a short communication, given that the manuscript is relatively short and concise (consider that to me, this is not a weakness or deficiency).

Still there are some improvements that you should consider to enhance the presentation and the discussion of the results.

Line 27: “95% CI 0.36-0.82” it is not clear what you are referring to.

Line 28: “95% CI 0.65-0.92” it is not clear what you are referring to.

Line 43: “Depending on iodine and fluid intake” sounds too vague, reformulate and/or specify better.

Line 72: “A sub-sample of 27 participants”. To me, this is not a sub-sample. This is actually your sample. Correct this in the entire manuscript.

Line 72: “Participants”. Do you mean women? Infants? Or both? Reformulate and/or specify better.

Line 73-74: Reformulate “All, except two of the woman recruited for the sub-sample study were breastfeeding”.

Line 181: In general, try to improve the discussion of data, referring to previous literature (see: Fernando, S. W., Cavedon, E., Nacamulli, D., Pozza, D., Ermolao, A., Zaccaria, M., ... & Mian, C. (2016). Iodine status from childhood to adulthood in females living in North-East Italy: iodine deficiency is still an issue. European journal of nutrition, 55(1), 335-340).

Line 184-185: “We found that the ICC of infant UIC had a poor to moderate agreement”. This is ok, but remember that you should discuss your data here. Therefore, try to provide an explanation, discussion or hypothesis for this statement.

Line 202: “The variability expressed by the ICCs and the mean CV, was somewhat higher for UIC than for BIC”. This is ok, but remember that you should discuss your data here. Therefore, try to provide an explanation, discussion or hypothesis for this statement. Try to compare your findings with previous studies investigating repeatability on milk Iodine measures (see: Niero, G., Franzoi, M., Vigolo, V., Penasa, M., Cassandro, M., Boselli, C., ... & De Marchi, M. (2019). Validation of a gold standard method for iodine quantification in raw and processed milk, and its variation in different dairy species. Journal of dairy science, 102(6), 4808-4815).

Figure 1: Maybe a graph with mean values and SD for each sample would be more informative and clear to present.

Figure 2: Maybe a graph with mean values and SD for each sample would be more informative and clear to present.

Reviewer 2 Report

This manuscript shows the reliability of Diaper collection method for the measurement of urine iodine. In this manuscript, there are several points which need to be addressed in order to strengthen the narrative.

The authors should mention if there are any other elements which are excreted in the urine similar to Iodine.  Please elaborate in the methods if repetitions were performed on each individual, i.e., if each individual only donated 2X5 ml sample once or if 2X5ml sample was extracted repeatedly for several days. Additionally, 2-spot sample for one individual does not seem enough to draw statistical significance. Several factors can influence the Iodine content on one given day. Atleast an average of 4-5 spots should be accounted for to establish the statistical relevance.   Since, ICP-MS was performed to measure the iodine content, the authors should presents the data for other relevant elements that can be used as a control. To ascertain that diaper collecting method is reliable, data from any other comparable method is not provided. Perhaps, the consistency of iodine measurements and reliablity of the method can be tested in adults and extended to the infants.  The discussion could use more elaboration and discuss why is 90% of I excreted and methods if any to improve Iodine's absorption in the gut. Minor: L188- the word should be stable Minor: L 151- the word should be breastfed.

Round 2

Reviewer 1 Report

The manuscript has been improved and is acceptable for publication. Still I suggest to change it into a short communication.

Reviewer 2 Report

The manuscript is acceptable in its current form.